# The Influence of Safety Culture and Climate on Safety Performance: Mediating Role of Employee Engagement in Manufacturing Enterprises in Ethiopia

Mesfin Abeje  and Fan Luo *

School of Management, Wuhan University of Technology, Wuhan 430070, China
* Correspondence: sailluof@126.com

**Abstract:** Manufacturing enterprises face a strategic challenge in managing risks, as safety concerns can result in huge costs for employee wellbeing and business success. However, there is no clear link between using diverse instruments to assess and measure safety performance and the culture and climate of safety in the workplace, which is likely due to differing perspectives on the topic. This study explores the influence of safety culture and climate on safety performance and on the mediating role of employee engagement in the Ethiopian manufacturing sector. This study was conducted using a quantitative research methodology 368, where three hundred and sixty-eight respondents from five large-scale industrial manufacturing enterprises were selected through purposive sampling. A combination of techniques was used, including structural equation modeling, growth paths, and correlation matrix, and these were performed using the SPSS/AMOS v. 24 software suites. These methods established a causal relationship between safety culture, safety climate, and safety performance. The study's main finding is that safety culture significantly impacts safety performance, which is followed by safety climate. Additionally, employee engagement played a significant mediating role between safety culture and safety performance, as well as between safety climate and safety performance. Based on these results, policymakers and practitioners in large-scale manufacturing enterprises in Ethiopia should prioritize improving the safety culture and climate of their workplaces to enhance safety performance and overall safety.

**Keywords:** safety culture; safety climate; safety performance; employee engagement

## 1. Introduction

"Occupational Health and safety are a concern for societal well-being as industrialization and the growth of the service industry is accelerating rapidly worldwide. Workplace health issues are on the rise. Risks to employee health and safety are being recognized as a driving construct in the hunt for answers to stop them from damaging the manufacturing industry workforce" [1]. In the safety literature, "safety climate" and "safety culture" refer to how businesses manage safety-related matters and how employees perceive and approach them. "Safety culture" represents an organization's commitment to the style and efficiency of safety and health management and is influenced by individual and group beliefs, attitudes, competencies, and performance patterns [2]. Experts assess the safety climate and employee perception of workplace safety policies, practices, and rules, which are now included in the definition of safety climate [3]. Over time, the definition of safety climate has been broadened to encompass this aspect as well [4]. The ongoing debate about safety climate parameters may vary across industries [5].

Safety culture and safety climate are distinct concepts, with safety culture referring to the fundamental principles guiding safety practices within an organization [6], and safety climate pertains to employees' perceptions and attitudes toward safety in their work environment. Both concepts are interconnected and important in promoting safety and reducing the risk of accidents and injuries. Safety culture is typically evaluated through

surveys or assessments, while employee surveys measure safety climate. By taking a comprehensive approach to safety that considers both safety culture and safety climate, organizations can create a safer work environment for their employees [7,8]. Due to the complexity of safety culture, safety climate is universally used as a snapshot of safety culture in research.

This study explores the intricate relationship between safety performance, safety culture, safety climate, and employee engagement in manufacturing enterprises in Ethiopia. This study visually represents this complex interplay in Figure 1 and highlights the mediating role of employee engagement in promoting a positive safety culture and climate to improve safety performance. Although research gaps persist in Ethiopia, this study has the potential to develop effective safety programs for sustainable economies and business success in manufacturing enterprises, particularly given the high rates of work-related injuries in developing countries such as Ethiopia [9–11].

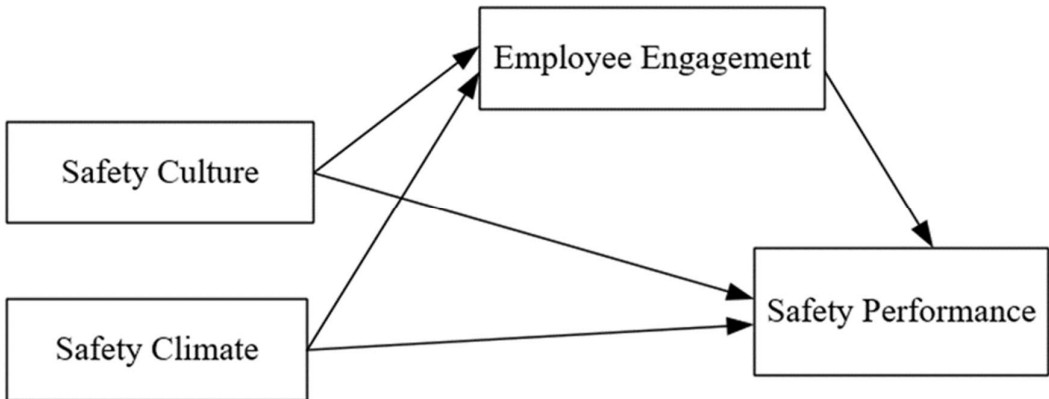

**Figure 1.** The proposed conceptual model of this study by the authors.

However, the organizational factors influencing safety culture, climate, employee engagement, and safety performance require further investigation [12]. Reports from the Ethiopian Ministry of Health reveal that among 16,611 large-scale industrial workers in Addis Ababa, 724 injuries per 1000 exposed workers occurred. Injury rates vary considerably in manufacturing enterprises, highlighting the critical importance of safety. Developing nations face a higher risk of work-related injuries, with rates 10 to 20 times higher than industrialized countries [13,14]. A study in Norway found 317 injuries per 1000 exposed workers over a year, while a U.S. study recorded 75 injuries per 1000 exposed workers per year among primary industrial workers [9,15,16]. These findings underscore the significant impact of safety issues in Ethiopia and the urgent need for effective safety programs in manufacturing enterprises [10,11]. The impact of safety issues on manufacturing enterprises creates obstacles and uncertainties, making workplace safety critical for sustainable business success.

Promoting a safe workplace is crucial for Ethiopian manufacturing businesses to attain their desired shift from an agricultural economy to industrialization. A safe work environment paves the way for an enterprise's progress, expansion, and prosperity. It leads us to the following questions: How do safety culture and climate impact the safety performance metrics of manufacturing enterprises? And how does employee engagement act as a mediator between these variables?

The concept of safety performance lacks a definitive definition in the literature. The relationship between safety culture, safety climate, employee engagement, and safety performance in manufacturing firms is a thriving research area, with varied publications exploring the topic. One conflicting hypothesis is whether safety culture and climate are distinct constructs [8,17]. While some studies argue that safety culture represents an organization's underlying values, beliefs, and assumptions about safety and that safety climate refers to employees' shared perceptions of safety, others believe that safety climate

manifests safety culture [3,7]. Another divergent hypothesis is the role of employee engagement in mediating the relationship between safety culture, climate, and performance [18]. While some studies have found a significant association between employee engagement and safety performance [19], others have produced mixed or inconclusive results [20]. Nowadays, millions of workers suffer from non-fatal accidents and job-related illnesses, with hazardous working conditions causing a shocking 7500 deaths daily [21]. It surpasses the annual deaths caused by traffic accidents, war, and violence. Inadequate knowledge and resources, unsustainable business practices, and a lack of a preventative culture contribute to this global issue. Sadly, governments and enterprises have failed to address this challenge, and economic liberalization and technological advancements exacerbate the problem, particularly in less developed nations [22,23].

Recent statistics from the Ministry of Labor and Social Affairs and Health Ethiopia reveal that manufacturing firms in Ethiopia are highly susceptible to occupational injuries, with 335 out of every 1000 workers being affected annually [11,24]. These injuries cause significant work absences, hospitalizations, and prolonged recovery periods, resulting in an average loss of 191 days [11,25]. Additionally, a study in Afar, Ethiopia, found that 783 out of 1000 employees suffered injuries, with 11% requiring hospitalization and 6153 days lost due to significant injuries [26]. These figures highlight the urgent need to prioritize workplace safety in Ethiopian manufacturing enterprises to ensure optimal performance. Nevertheless, the current study contributes to the field by exploring these variables' influences within a specific context.

This research investigates the influence of safety culture and climate on safety performance in Ethiopian manufacturing companies with the mediating effect of employee engagement. While previous studies have explored the relationship between safety culture, climate, and safety performance [18,19], none have examined their unique impact on the mediating effect of employee engagement in Ethiopia's large-scale manufacturing enterprises [26]. A safe working environment is essential for employee wellbeing and the sustainability of manufacturing enterprises, protecting employees and ensuring that the company can operate sustainably by preventing accidents, injuries, and equipment damage. This study's contribution is vital for policymakers, investors, and practitioners in developing countries, validating the social cognitive theory in the Ethiopian context and emphasizing the importance of safety performance in the manufacturing industry throughout Africa.

## 2. Literature Review and Hypothesis Development

The aim was to collect and assess safety-focused research based on empirical data from 95% of articles published between 2000 and 2022, as well as 47% of source journals published between 2017 and 2022. A wide range of unique keywords were used throughout the study. The keywords were manufacturing, safety climate, safety culture, employee engagement, and safety performance. The following keywords were used to search reputable databases and indexing services, such as Google Scholar, Web of Science, and Engineering Village.

### 2.1. Safety Culture

The term "culture" pertains to companies, safety, and workplace safety. By presenting relevant empirical data and their theoretical advances, this paper offers some clarification on positive safety culture, levels of aggregation, and safety performance. Safety culture frequently impacts employees' attitudes and conduct toward an organization's ongoing health and safety performance [8]. Several studies have explored how safety culture could impact safety performance, especially in high-risk enterprises [27–29]. However, enterprises are still in the early stages of developing safety performance. Cooper argued that special treatment is a safety culture. The progress, nature, and competence of safety management are subjective to the values, attitudes, perceptions, and competencies of an individual and a group and the social behaviors that make up an organization's safety culture. In

addition, Zohar argues that this structure makes sense because an employer is responsible for their legal responsibility. Just as a successful quality system requires participation from the entire workforce, so does a positive culture of safety and wellbeing. A shared commitment in terms of attitudes and values is necessary. Workers must have confidence that safety protocols are effective and committed to, regardless of whether this means missing financial and performance goals [30]. In contrast, the field of culture recognizes that simply implementing culture programs is not enough to create lasting change. Instead, they advocate for a comprehensive and practical approach to exploring and understanding organizational culture for real progress to occur [31–33]. Schein stresses the importance of understanding culture to manage it effectively. He warns that simplifying the approach to culture management can create a mistaken perception of influencing culture while failing to achieve efficiencies. Cooke recognizes that organizational culture can generate significant influence and is becoming recognized globally across all industries. Simon emphasizes the importance of promoting safety cultures as an essential technique for improving safety performance, fostering trust, employee engagement, and psychological safety. This study's main objective is to ascertain how safety culture impacts safety performance and to examine employee engagement's role as a mediator. These variables include context, management of safety risk, and information-sharing techniques. Safety culture frequently affects employees' attitudes and conduct toward an organization's ongoing health and safety performance [19].

*2.2. Safety Climate*

Zohar was the first to introduce the term "safety climate" in the literature, defining it as "a summary of micro-level perceptions that employees hold regarding their work environments". Safety climate represents a crucial intersection between organizational and psychological processes and their relationship with safety. A growing body of research indicates that the safety climate of an organization affects employees' desire for safety-related work and that this motivation affects individual practices and organizational safety outcomes [29]. The safety climate in an organization significantly influences employee motivation to engage in safety-related work, which in turn impacts individual conduct and organizational safety outcomes. Research shows that safety climate is positively related to safety motivation, compliance, and participation behaviors, leading to improved safety outcomes such as lower injury rates and near-miss incidents. Safety training, open communication, involving employees in safety decision-making, and recognizing safe behaviors can create a positive safety climate. By prioritizing safety and creating a positive safety climate, organizations can improve safety performance and protect their employees' wellbeing [34–36]. The definition, measurement, and application of the safety climate construct to enhance safety in manufacturing have not yet been thoroughly examined in any studies in the study area, particularly in Ethiopia; however, in the survey by Zohar, several conceptual problems with the safety climate that need to be addressed to further advance the theory of the safety climate emerged due to conceptual ambiguity, which encouraged a shift from practical to more theoretical difficulties. Increasingly, companies have become engaged in using safety climate surveys to examine an organization's safety performance.

Both academically and commercially, researchers have shown that an organizational safety climate significantly impacts risky manufacturing practices in various industrial contexts, including chemical and nuclear processing [37]. A safety climate is when firms follow safety procedures, norms, and rules; these form a safety climate. Y. Chena argued that a safety climate directly impacts safety performance. It has been noted that various settings' safety climates substantially impact other contexts' safety performance. Jiang and Lixin's meta-analysis contends that there is a significant link between safety climate and safety performance. Various authors have argued that this linkage leads to improved workplace safety outcomes and safer practices [38] and reduced injury rates [39,40]; furthermore, the study conducted by Christian surveyed employees from diverse sectors to explore the effect between safety climate and safety performance [35]. The outcomes indicated that safety

climate and employee engagement were positively correlated with safety performance. Several studies have demonstrated a notable association between safety performance and safety climate [41–43]. As opposed to Beus et al. argued that a safety climate reduces accident rates. Nevertheless, research has disputed this strong association and has claimed that a safety climate has little bearing on safety performance [44]. Reliable and compelling methods are currently confusing and have heightened safety. Their relationship with the safety climate still needs to be examined. The literature, however, argues that a booming safety climate positively impacts safety performance through an efficient safety climate. And the authors consider that a study's context, such as its location and industry, are significant variables that could impact its results [45].

*2.3. Safety Performance*

Safety performance in large-scale manufacturing enterprises refers to the effectiveness of safety management practices and programs in preventing workplace accidents and injuries [46,47]. There are several debates surrounding safety performance in large-scale manufacturing enterprises, which reflect the complex and dynamic nature of this field. The issue is widely debated, and research has established that safety climate is a significant predictor of safety performance across multiple industries, such as manufacturing, construction, healthcare, and transportation. For instance, a study by Hofmann and Stetzer demonstrated that safety climate significantly predicted injury rates in a manufacturing facility. The study's authors recommend that organizations can boost their safety performance, safeguard their employees' wellbeing, and enhance their image as secure and responsible employers by fostering a positive safety climate.

In contrast, Guldenmund argue that safety performance in large-scale manufacturing enterprises is closely associated with organizational culture, which shapes the attitudes and behaviors of employees toward safety [18]. The authors suggest that safety management should develop a positive safety culture to improve performance. Furthermore, safety performance in large-scale manufacturing enterprises is influenced by the interaction between safety management practices and worker practices. The authors argue that safety management should focus on promoting safe deeds among workers, as well as implementing effective safety management practices to enhance safety performance. The argument put forth by Chena, Y is that that safety climate directly influences safety performance and has important implications for organizations. The authors suggest prioritizing safety and creating a positive safety climate. Organizations can improve safety performance, protect the wellbeing of their employees, and enhance their reputation as safe and responsible employers. So, these debates highlight the complexity of safety performance in large-scale manufacturing enterprises and the need for effective safety management practices considering various internal and external factors [38].

Generally, the literature has evidenced the impact of the proposed variable on safety performance. The present research investigated the effect between safety culture, safety climate, and the mediating function of employee engagement on safety performance in the manufacturing sector. These have bridged the gap between the importance of safety performance in large-scale manufacturing enterprises in Ethiopia and have given insight into effective safety management practices to prevent workplace accidents and injuries.

*2.4. The Impact of Safety Culture and Climate on Safety Performance*

2.4.1. The Relationship between Safety Culture and Safety Performance

The concept of safety culture, according to Yorio et al., was introduced following the Chernobyl event in 1986; however, no one definition of safety culture has been agreed upon by all researchers since its conception [48]. At the same time, various researchers have found that catastrophes or damages strongly influence the safety culture of an organization [49] and have concluded that safety culture and near misses harm safety performance. Numerous investigations have explored the connection between safety culture and safety performance in manufacturing enterprises [50], and one study investigated the relationship



between safety culture and safety performance in a large manufacturing firm in China. The investigation showed that safety culture substantially affects safety performance, with safety culture having a more significant impact on safety performance. The study suggests that a good safety culture can encourage a proactive approach to safety performance and lower the likelihood of workplace accidents and injuries in large manufacturing firms. As mentioned, safety culture refers to the shared values, attitudes, beliefs, and behaviors related to safety within an organization. Safety performance, on the other hand, refers to the efficiency of safety management practices and programs in preventing workplace accidents and injuries. Studies have consistently shown that safety culture is strongly influenced by safety performance in large-scale manufacturing enterprises. A positive safety culture, which is characterized by management being interested in safety, communication, participation, and safety training and education, is associated with better safety outcomes, including fewer accident rates and injuries. Nevertheless, several debates exist regarding the association between safety culture and safety performance in large manufacturing companies.

Some references that discuss these debates demonstrate that safety culture is a complex and multifaceted concept that is challenging to measure and define. The authors suggest that safety culture should be assessed using multiple indicators, such as employee perceptions of safety, safety-related behaviors, and safety outcomes [51]. Conversely, Flin and Mearns contend that the relationship between safety culture and safety performance is contingent on the context. The authors suggest that safety culture should be evaluated in the context of specific organizational and industry factors, such as the nature of the work, the level of risk, and the regulatory environment [52]. Additionally, Lawton et al. contend that the caliber of safety management practices affects the correlation between safety culture and safety performance. The authors propose that efficient safety management practices are crucial for fostering a good safety culture and enhancing safety performance [2].

Moreover, employee attitudes and behaviors mediate the relationship between safety culture and safety performance. The writers propose that an organization's safety culture shapes its employees' safety perceptions and willingness to participate in safe practices, thus impacting safety performance. They argue that employee attitudes and behaviors mediate the relationship between safety culture and safety performance.

The authors suggest that safety culture shapes employee perceptions of safety and their willingness to engage in safe practices, thus influencing safety performance [53]. These debates highlight the complex and multifaceted relationship between safety culture and safety performance in large-scale manufacturing enterprises. While there is general agreement that safety culture is an essential predictor of safety performance, the specific nature of this relationship is likely to vary depending on a range of organizational and contextual factors. However, there is limited research on the relationship between safety culture and safety performance in large Ethiopian manufacturing firms within the Ethiopian context. Still, some studies have identified gaps and challenges in this field. One study found a lack of awareness and understanding of safety culture and its importance in promoting safety performance. The study also identified several challenges in developing and maintaining a positive safety culture, including limited resources, inadequate training, and a lack of regulatory enforcement [54], so this study highlights the need for further research into the relationship between safety culture and safety performance in large-scale manufacturing enterprises in Ethiopia. There is a need to develop an influential safety culture to improve safety performance in the manufacturing sector. So, we propose our hypothesis, and we want to test it.

**Hypothesis 1:** *Safety culture positively influences safety performance in Ethiopian large-scale manufacturing enterprises.*

2.4.2. The Relationship between Safety Climate and Safety Performance

Occupational health and safety place significant emphasis on safety climate and safety performance as crucial concepts. Safety climate pertains to the collective perceptions, attitudes, and beliefs of employees regarding safety within an organization. On the other hand, safety performance pertains to the tangible safety outcomes that an organization attains. One study has shown a strong relationship between safety climate and safety performance in large manufacturing companies. The study examined the relationship between safety climate and safety performance in a large-scale manufacturing enterprise in Israel [55]. The study revealed a significant relationship between employees' perceptions of safety climate and the frequency and severity of workplace accidents. More specifically, a positive safety climate was associated with reduced accidents and fewer severe injuries [55]. Another study explored the relationship between safety climate and safety performance in a large-scale manufacturing enterprise in Australia. The study found that a positive safety climate resulted in higher levels of safety performance, as measured by the number of accidents and injuries [56].

These studies show how safety climate significantly predicts safety performance in large manufacturing firms, but there are debates surrounding measuring safety climate and safety performance. This suggests that it is essential to consider various factors, such as mediating variables and context. Employee engagement has been found to mediate the relationship between safety climate and safety performance, and the effect differs depending on the industry and organizational culture. Manufacturing firms must prioritize safety to create a safer work environment and improve safety performance. This involves considering the safety climate and promoting a positive safety climate [17,41].

The inconsistencies in the existing literature provide enough evidence to conduct further research on safety climate. The current study will contribute to the conceptual understanding framework by uncovering the disparities between the existing notions of safety climate and safety performance. This investigation focuses on the impact of safety climate on safety performance in large Ethiopian manufacturing firms. Therefore, based on this reasoning, we propose our hypothesis:

**Hypothesis 2:** *Safety climate positively influences safety performance in Ethiopian large-scale manufacturing enterprises.*

*2.5. Mediating Role of Employee Engagement*

2.5.1. Safety Culture, Employee Engagement, and Safety Performance

Multiple sectors, including manufacturing, have demonstrated that employee engagement mediates the relationship between safety culture and performance. Employee engagement refers to how much employees are emotionally invested in their work and their organization. Engaged employees are more important in exhibiting good safety-related conduct and in establishing a positive safety culture, resulting in improved safety performance.

Several research investigations have explored the impact of employee engagement as a mediator in the relationship between safety culture and safety performance. One study evaluated 305 Chinese manufacturing firms and found that employee engagement mediated the effect between safety culture and safety performance [19]. The same authors, furthermore, studied 342 workers in a large-scale manufacturing enterprise in China. They found that employee engagement fully mediated the relationship between safety culture and safety performance [57]. While there is a consensus that employee engagement mediates the relationship between safety culture and safety performance in large-scale manufacturing enterprises, in contrast, there are some debates and criticisms surrounding this relationship. There are a few references that discuss these debates; Clarke and Ward argue that the relationship between safety culture, employee engagement, and safety performance may be affected by contextual factors such as organizational size and industry

type. The authors suggest that future research should consider these contextual factors when examining the relationship between these variables.

Furthermore, the influence between safety culture, employee engagement, and safety performance is intricate and may vary based on the specific dimensions of each variable. The authors suggest that future research should examine the dimensions of safety culture, employee engagement, and safety performance that are most relevant to each organization. While there are some debates and criticisms surrounding the mediating role of employee engagement on the relationship between safety culture and safety performance in large-scale manufacturing enterprises [17], some criticize this perspective and propose alternative models. Critics argue that first, other variables like safety climate may be more direct drivers of safety performance. Second, safety culture may be directly related to safety performance without employee engagement as a mediator. Third, how employee engagement is defined and measured in this context may be inappropriate. Though evidence still supports employee engagement as a mediator, the relationship may be more direct. Other variables likely also play a role. The relationship between safety culture, employee engagement, and safety performance is complex with various interconnections. While employee engagement may mediate the link between safety culture and safety performance, alternative models and additional variables should be considered. A balanced perspective acknowledging these complexities is needed [35,36,58]. While there is a lack of research specifically on the mediating impact of employee engagement on the relationship between safety culture and safety performance in large-scale manufacturing enterprises in Ethiopia [26], additional research is necessary to investigate the particular function of employee engagement as a mediator in this context.

**Hypothesis 3:** *Employee engagement has a significant positive mediation impact on the relationship between safety culture and safety performance.*

2.5.2. Safety Climate, Employee Engagement, and Safety Performance

Numerous investigations have demonstrated the mediating role of employee engagement in the relationship between safety climate and safety performance. Safety climate pertains to the collective perceptions and attitudes of employees towards safety in the workplace. A favorable safety climate encourages safe practices and reduces the probability of accidents and injuries. Huang et al.'s research found that safety communication and climate positively affected worker trust and engagement in safety, which, in turn, was linked to enhanced safety performance. The writers propose that safety communication and climate can directly affect safety performance, but the relationship is partly mediated by employee engagement [59].

Likewise, a research investigation discovered that safety climate significantly impacted employee practices and safety performance. The study found that safety climate was positively correlated with employee safety conduct, and employee attitudes and perceptions directly mediated this relationship. Employee engagement is a crucial mediating effect between safety climate and performance [19]. Clarke's research investigation revealed that safety climate had a direct and substantial impact on safety performance, but employee attitudes and behaviors partially mediated this relationship [41]. The writers propose that employee engagement is a crucial mechanism through which safety climate can impact safety performance. These studies highlight the significance of employee engagement in the correlation between safety climate and safety performance. By promoting a constructive safety climate and encouraging employee engagement, organizations can enhance safety performance and lower the likelihood of accidents and injuries. These findings have supported the mediating impact of employee engagement on safety climate and safety performance; there are also some debates and challenges in the literature. One central discussion concerns the definition and measurement of employee engagement. Employee engagement is a multifaceted concept encompassing different aspects, such as job satisfaction, motivation, and organizational commitment. There is still some disagreement

among researchers about how to define and measure employee engagement and how to distinguish it from related constructs. Another debate is around the direction of causality in the relationship between safety climate, employee engagement, and safety performance. Although numerous investigations propose that a sympathetic safety climate increases employee engagement and leads to improved safety performance, some researchers contend that the relationship may be bidirectional or even reversed. For instance, a research investigation discovered that safety performance was positively correlated with safety climate, and employee attitudes and behaviors partially mediated this relationship [60,61]. Despite research in other contexts supporting employee engagement as a mediator in the correlation between safety climate and safety performance, additional investigation is necessary. At the same time, there is a lack of research specifically on the mediating impact of employee engagement on the relationship between safety climate and safety performance in large-scale manufacturing enterprises in Ethiopia, Additional research is necessary to investigate the particular function of employee engagement as a mediator in this context, particularly within the Ethiopian context [62,63].

**Hypothesis 4:** *Employee engagement has a significant positive mediation influence on the relationship between safety climate and safety performance.*

### 2.5.3. Employee Engagement and Safety Performance

Safety performance is crucial for the success of a business as it ensures the prevention of significant incidents and the management of safety risks. Numerous investigations have explored the correlation between employee engagement and safety performance in large manufacturing firms. Employee engagement pertains to the extent of emotional investment and commitment that employees have towards their work and their organization. Safety performance refers to the actual safety outcomes achieved by the company, such as the number of accidents. Probst examined the relationship between employee engagement and safety performance in a large manufacturing enterprise [64]. The study found that employee engagement was positively correlated with safety performance, even when considering other variables such as safety climate and culture. The investigation proposed that engaged employees are more likely to display good safety behaviors like complying with safety procedures, reporting hazards, and participating in safety training.

Likewise, a research investigation discovered that employee engagement partially mediated the relationship between safety climate and safety performance in a large manufacturing firm in Taiwan. The study proposed that encouraging a safety climate can increase employee engagement, leading to enhanced safety performance [65]. Furthermore, the researchers discovered that employee engagement partially mediated the correlation between safety leadership and safety performance in a large manufacturing firm. The study proposed that efficient safety leadership can result in increased levels of employee engagement, leading to enhanced safety performance [66]. Moreover, Burke and colleagues investigated the correlation between employee engagement and safety performance in a large American manufacturing firm. The study revealed that employee engagement was positively correlated with safety compliance and participation, even after adjusting for other variables such as safety climate and culture. The study proposed that engaged employees are more inclined to adhere to safety procedures and play an active role in recognizing and reporting safety hazards [35]. While evidence suggests a positive relationship between employee engagement and safety performance in large-scale manufacturing enterprises, there are also debates and criticisms surrounding this topic. One debate concerns the potential for engagement initiatives to be used as a managerial control or manipulation method. Engagement initiatives and programs are viewed as a way for organizations to extract a more discretionary effort from employees while masking the underlying issues with workplace safety. Therefore, some researchers suggest the need for a more critical and nuanced approach to understanding the relationship between engagement and safety [67].

Additionally, some researchers have raised concerns about the generalizability of findings to different types of organizations and industries. The relationship between engagement and safety performance may vary depending on organizational culture, industry regulations, and job demands. Therefore, some researchers suggest more context-specific research to better understand the relationship between engagement and safety in different contexts [68]. In general, although evidence supports a beneficial correlation between employee engagement and safety performance in large manufacturing firms, there are still debates and criticisms regarding this subject. Therefore, future research should continue to investigate these concerns and offer a more holistic understanding of the relationship between employee engagement and workplace safety.

**Hypothesis 5:** *Employee engagement has a substantial positive impact on safety performances.*

### 2.6. Conceptual Framework for the Research

Following the literature review, the authors of this study developed a conceptual research model for their study. Notably, there have been no comparable studies on business performance in Ethiopia, and this innovative contextualized approach focused on employee engagement as the mediating variable with safety performance as the outcome variable. The predictive factors included safety climate and safety culture. As such, the conceptual framework of the derived model sought to validate and estimate the model's fitness while also supporting the hypothesized direct and indirect effects of the mediating variable, employee engagement. To this end, the proposed conceptual research model employed the assumptions of structural equation modeling, path modeling, and growth path modeling analysis to bolster the study's findings, as illustrated in Figure 1.

## 3. Research Methodology
### 3.1. The Study Design

The purpose of a research design is to attain research objectives and address research questions. The researchers adopted a quantitative study approach. The study examined the influence of safety climate and safety culture on safety performance and how employee engagement mediates this relationship. While previous research has focused on the effect of safety climate and culture on performance, this research examined the mediating role of employee engagement. While previous studies have been limited to specific industries, this research explored these relationships across Ethiopian manufacturing firms [62,63]. Here, it disregarded the effects of other factors, such as the mediating role of employee engagement. Therefore, this study examined how safety culture and safety climate affect safety performance by mediating employee engagement in Ethiopia's large-scale manufacturing enterprises.

### 3.2. Sampling Techniques and Data Collection

This study utilized a population of 1055 employees to determine the sample size, employing purposive sampling techniques to select 368 respondents [69]. Furthermore, the study determined a sample size of 5 manufacturing companies from a pool of 15, resulting in a cluster of participants and companies that enhanced the quality of the quantitative research methods. For the selection of individual respondents, simple random-sampling techniques were used to ensure unbiased and equal representation. The researchers distributed 385 questionnaires to respondents. Finally, 368 questionnaires were collected and analyzed using the SPSS/AMOS statistical tools. This approach ensured that data were collected from a diverse and representative group of participants.

### 3.3. Measurement Items

This study utilized meticulously designed questionnaires that were carefully adapted from prior research to measure the various factors at play. A five-point Likert scale was used to assess the dependent, independent, and mediating variables, resulting in twenty-four

statement assessment items and four construct variables. This comprehensive approach facilitated a nuanced understanding of the subject matter.

To evaluate safety performance, which was the dependent variable in the analysis, established measuring methods were used that underwent modifications based on previous findings [66,70]. Additionally, six contextualized items were evaluated using pre-established five-point Likert scales. This study delved into workplace safety climate and culture [8,34,71], ultimately identifying employee engagement as the final mediating variable [71,72], showed the details in the Appendix A. Overall, the results and extrapolations of this study were based on a rigorous and thorough methodology, yielding valuable insights into the complex interplay of the factors at play in the subject matter.

## 4. Results and Interpretations

### 4.1. Demographic Data Analyses

The demographic data for this study on safety culture, safety climate, employee engagement, and safety performance revealed that the sample had a near-equal distribution of male and female respondents, with the largest age group being respondents between 31–35 years old. The high proportion of respondents with a bachelor's degree suggested a potential bias towards individuals with higher levels of education, and the diverse range of work experience suggested relevance to individuals with different levels of experience. It was essential to consider these factors in analyzing and interpreting this study's results to avoid limitations or biases.

### 4.2. Evaluation of the Measurement Model

This study evaluated the accuracy of the measurement model by assessing the construct validity and reliability of the items using convergence and discriminant validity [73]. What proportion of the standard deviation can be explained by items that measure the same construct and exhibit convergent validity [74]? If the sum of the average variances extracted (AVEs) is less than zero, the structures can account for more measurement errors than the variance (zero). However, it is not mathematically possible for the AVE to be less than zero, so this statement may contain an error. In this study, the AVE for each variable was more significant than 0.50, indicating that it could be considered a crucial indicator.

#### 4.2.1. Discriminate Validity Test and Reliability Test

The researchers employed the SPSS/AMOS 23 statistical software. We used this to evaluate the validity and reliability of all the relevant questions in the questionnaire used in this study. Various tests—validity, reliability, discriminant validity, and multidimensionality checks—were used to validate and assess the model fit indices. The results were estimated based on the average variance, loading factor, composite reliability (C.R.), and Cronbach's alpha values. Cronbach's alpha is a measure of internal consistency, which indicates how closely related a set of items are as a group. It ranges from 0 to 1, with higher values indicating greater internal consistency. A commonly accepted threshold for Cronbach's alpha is 0.70 or higher, meaning good reliability. This study's result showed that the Cronbach's alpha values for each variable ranged from 0.902 to 0.967, suggesting a high level of internal consistency across all variables.

Factor analysis identifies latent variables that may account for covariance among observations; the factor loading values measure the extent to which a variable can account for the variance in a particular factor, with a factor loading of 0.70 being considered significant. The estimated factor loading values fell between 0.701 and 0.877, indicating a high level of convergent validity for the measurement constructs across all the variables.

Furthermore, all the estimated critical ratio (C.R.) values for the latent constructs, which ranged from 0.924 to 0.984, surpassed the acceptable level of reliability for a critical ratio of 0.70 with a *p*-value of 0.05. Additionally, a regressed weighted essential balance greater than 1.96 was considered a significant parameter [75]. Because the level of AVE exceeded the suggested value of 0.50, it was regarded as excellent and acceptable [73];

for the entire derived construct, the estimated (AVE) values ranged from 0.671 to 0.912. In addition, the AVE value showed the substantial influence of the latent variable on the relationship between the predictive and outcome variables. This suggests that the measurement questions used in the conceptual model for the study, as shown in Table 1, may more accurately reflect the characteristics of each research variable, as presented in Table 1.

**Table 1.** Demographic information.

| Demographic Variables | | Frequency | % |
|---|---|---|---|
| Gender | Male | 186 | 50.4 |
| | Female | 182 | 49.6 |
| Age | 20–25 | 39 | 10.5 |
| | 26–30 | 47 | 12.8 |
| | 31–35 | 109 | 29.7 |
| | 36–40 | 101 | 27.4 |
| | Above 40 | 72 | 19.6 |
| Education | Diploma | 71 | 19.2 |
| | Bachelor's | 103 | 27.9 |
| | Master's | 97 | 26.3 |
| | Doctor | 8 | 2.1 |
| | Others | 89 | 24.1 |
| Experience (years) | 1–5 | 91 | 24.7 |
| | 6–10 | 97 | 26.3 |
| | 11–15 | 74 | 20.1 |
| | Above 15 | 106 | 28.8 |

Source: authors' computation.

### 4.2.2. The Correlation Matrix and Discriminate Validity

The constructs and the end variable, safety performance, safety climate, safety culture, and employee engagement, had a significant association based on the covariance values. There were strong and significant correlations between the independent and dependent variables, as the AVE from the observable variables, with a *p*-value of 0.05, demonstrates. The covariance values, therefore, showed that all the constructs were interconnected. Moreover, safety performance (S.P.), safety climate (SCL), and safety culture (S.L) were also interconnected. The mediating variable is revealed in Table 2. Employee engagement (EEG) also supported the correlations between the variables.

**Table 2.** Analysis of the reliability and validity tests.

| Constructs | Items | Factor Loading | Cronbach's Alpha | Composite Reliability | The Average Variance Extracted (AVE) |
|---|---|---|---|---|---|
| Safety performance | SP1 | 0.877 | | | |
| | SP2 | 0.936 | | | |
| | SP3 | 0.935 | 0.967 | 0.973 | 0.858 |
| | SP4 | 0.952 | | | |
| | SP5 | 0.936 | | | |
| | SP6 | 0.919 | | | |

**Table 2.** *Cont.*

| Constructs | Items | Factor Loading | Cronbach's Alpha | Composite Reliability | The Average Variance Extracted (AVE) |
|---|---|---|---|---|---|
| Safety culture | SC1 | 0.947 | | | |
| | SC2 | 0.984 | | | |
| | SC3 | 0.95 | 0.981 | 0.984 | 0.912 |
| | SC4 | 0.916 | | | |
| | SC5 | 0.969 | | | |
| | SC6 | 0.964 | | | |
| Safety climate | SCL1 | 0.921 | | | |
| | SCL2 | 0.92 | | | |
| | SCL3 | 0.953 | 0.957 | 0.965 | 0.823 |
| | SCL4 | 0.876 | | | |
| | SCL5 | 0.943 | | | |
| | SCL6 | 0.824 | | | |
| Employee engagement | EEG1 | 0.781 | | | |
| | EEG2 | 0.886 | | | |
| | EEG3 | 0.896 | 0.902 | 0.924 | 0.671 |
| | EEG4 | 0.884 | | | |
| | EEG5 | 0.753 | | | |
| | EEG6 | 0.701 | | | |

Note: SCL = safety climate; SC = safety culture; EEG = employee engagement; SP = safety performance.

### 4.3. Analysis of Multiple Regressions

The responsive components that impacted the safety performance results were identified using multiple regression analysis in this study. A hugely important and beneficial relationship existed between safety culture, safety climate employee engagement, and safety performance. The estimated value of each parameter, according to the regression results, ranged from 0.229 (EEG) and 0.236 (SCL) to 0.511 (S.C.) with (S.P.). The mediating role of employee engagement elucidated the significant and positive indirect impact among the variables, with the estimated parameter values ranging from 0.402 (SCL) to 0.468 (S.C) when employee engagement (EEG) was considered. The critical ratio (C.R.) values were derived by dividing the estimates by their corresponding standard errors (S.E.) and assessing the results. If the critical ratio score is above 1.96, there is a higher probability that the impact will be positive at a significance level of 0.05. The statistics indicated that every research constraint was statistically significant at a *p*-value of 0.05.

Each variable in the proposed research model was subjected to individual testing to ensure it met the assumptions stated earlier in the article. This ensured that the proposed model was based on reliable and valid measurements of the studied variables. The predictor variables, safety climate (SCL) and safety culture (S.C.), positively and substantially impacted the outcome variable, safety performance, with a *p*-value of 0.000. Also, the mediator variable, employee engagement (EEG), significantly and constructively influenced the dependent variable, safety performance (S.P.), with a *p*-value of 0.000. This outcome suggested that the study variables' variances across all dimensions were similar. Therefore, employee engagement (EEG) was more likely to significantly impact safety performance (S.P.), as shown in Table 3 below.

**Table 3.** Correlations matrix and discriminate validity results analysis.

|  | SP | SC | SCL | EEG |
|---|---|---|---|---|
| SP | 0.926 | | | |
| SC | 0.901 | 0.955 | | |
| SCL | 0.858 | 0.857 | 0.907 | |
| EEG | 0.834 | 0.812 | 0.803 | 0.819 |

Note: SCL = safety climate; SC =safety culture; EEG = employee engagement; SP = safety performance.

### 4.4. Structural Equation Model Estimate

In this study, the researchers used SEM to test the impact of one or more response variables on one or more dependent variables. The path diagram helped to visualize the relationships between these variables and to identify any direct or indirect effects of the response variables on the dependent variables. The estimation of the structural equation model showed that the goodness-of-fit indices were considered acceptable if they surpassed the underlying assumptions' threshold values. However, the reference for these threshold values was not provided by the user. Furthermore, the chi-square value for the model was 1.74, which was below 3.0, indicating a good fit [76].

Additionally, the goodness-of-fit index (GFI) was 0.932, exceeding the acceptable threshold of 0.90. The incremental fit index (IFI) was 0.942, surpassing the recommended value of 0.90, and the adjusted goodness-of-fit index (AGFI) was acceptable at 0.916, exceeding the suggested value of 0.90. The normed fit index (NFI) was also above 0.90, i.e., 0.902. The comparative fit index (CFI) for the model was 0.961/0.961, exceeding the threshold value of 0.90, and the Tucker-Lewis index (TLI) was 0.901. Therefore, Table 4 provides a summary of the model fit indices.

**Table 4.** Regression weights for significant and critical ratio levels.

| Variables | Path | Measurements | ($\beta$) | S.E. | C.R. | Sig. |
|---|---|---|---|---|---|---|
| EGG | <--- | Safety culture | 0.468 | 0.072 | 6.512 | 0.000 |
| EGG | <--- | Safety climate | 0.402 | 0.069 | 5.83 | 0.000 |
| SP | <--- | Employee engagement | 0.229 | 0.045 | 5.052 | 0.000 |
| SP | <--- | Safety culture | 0.511 | 0.048 | 10.631 | 0.000 |
| SP | <--- | Safety climate | 0.236 | 0.043 | 5.445 | 0.000 |

Note: C.R. = critical Ratio and S.E. = standard error; EEG = employee engagement; SP = safety performance.

### 4.5. Results of Path Modeling and Goodness-of-Fit Analysis of the Structural Model

This study employed structural equations and growth path models to evaluate the goodness-of-fit and the effects of the constructs on the corresponding variables. Additionally, this study also considered the explained variance of the model fitness, which was based on the results of the path modeling diagram analysis and the estimated values of the growth path modeling coefficients or standardized regression weights. A value of 0.70 (70%), represented by $R^2$, was considered in the analysis. According to the growth path model, the independent variable, safety climate ($\beta$ = 0.236 ***), was statistically significant with a *p*-value of 0.000. Based on the estimated values of the standardized coefficient regression, this study concluded that the model was a good fit, as the R-squared ($R^2$) value was considered to be 0.85 (85%), and the *p*-value was significant at 0.000. As shown in Figure 2, the variable safety culture ($\beta$ = 0.511 ***) significantly impacted the outcome variable, safety performance, which was significantly mediated by employee engagement ($\beta$ = 0.229 ***). As a result, there was a high correlation between employee engagement's role in supporting safety performance in large-scale manufacturing enterprises in Ethiopia.

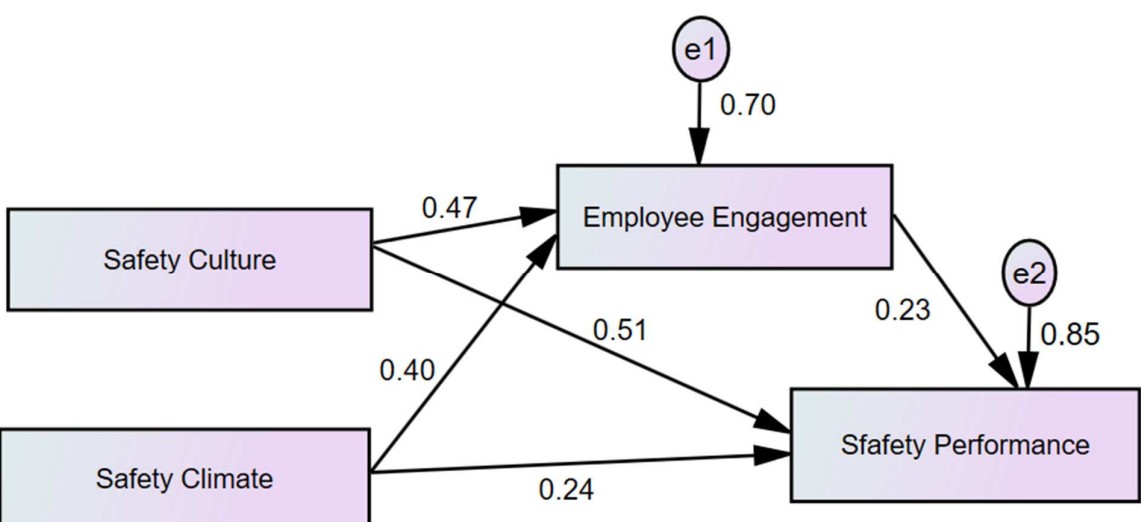

**Figure 2.** The structural equation model and significance of standardized path coefficients.

In contrast, the effect of each constraint on safety performance was magnified, while that of others remained unchanged. The outcome variable, safety performance (S.P.), was estimated to increase by a factor of β = 0.236 when the impact of safety climate (SCL) increased by one unit. This increase was statistically significant with a *p*-value of 0.000. The estimated value for safety culture's impact on the outcome variable, safety performance (S.P.), was found to be β = 0.511, which was statistically significant with a *p*-value of 0.000, indicating a considerable improvement in safety performance. Furthermore, the estimated factor of β = 0.229 with a *p*-value of 0.000 was likely to significantly impact the overall safety performance of large-scale manufacturing enterprises as it strengthened the mediating effect of employee engagement on the other variables such as safety culture and safety climate. Employee engagement mediated the relationship between the independent and dependent variables, as shown by the suggested conceptual model fit in Figure 2.

### 4.6. The Direct, Indirect, and Total Impacts of the Mediating Variable

This study used the growth path model to explore how the studied variable affected mediation. It aggregated the detrimental effects of the direct, indirect, and mediating variables on the support, as evaluated using path modeling regression weight analysis. The structural equation model (SEM) accounted for the mediating effects of the outcome variable and balanced all the independent factors. The results of the mediating variable illustrated the combined, direct, and indirect impacts of the mediating variable on safety performance. The outcomes of the growth path modeling demonstrated that the mediation weights considered two perspectives. The first investigated the mediating effects of employee engagement and the consequences of each independent element. Employee engagement had a positive and statistically significant mediation impact on safety culture and safety climate at β = 0.468 and β = 0.402, respectively. This result implied that employee engagement significantly influenced safety culture and climate. The second aspect that required examination was how employee engagement affected safety performance. Employee engagement positively and significantly influenced safety performance, encouraging the total effects (β = 0.229). In conclusion, all the direct, indirect, and total effects measurements showed that the mediating impacts of employee engagement and all the independent determinants on safety performance were highly significant and constructive. Thus, as demonstrated in Table 5 below, the role of employee engagement was advantageous for manufacturing enterprises considering safety culture and safety climate.

**Table 5.** The total, indirect, and direct effects.

| | SC | | | SCL | | | EEG | | |
|---|---|---|---|---|---|---|---|---|---|
| Variable | DE | IE | TE | DE | IE | TE | DE | IE | TE |
| EEG | 0.468 | 0.000 | 0.68 | 0.402 | 0.000 | 0.402 | 0.000 | 0.000 | 0.000 |
| SP | 0.511 | 0.107 | 0.618 | 0.236 | 0.092 | 0.328 | 0.229 | 0.000 | 0.229 |

Note: SCL= safety climate; SC= safety culture; EEG = employee engagement; SP = safety performance, DE = Direct effects; IE = Indirect effect; TE = total effect.

*4.7. Mediating Effects of Employee Engagement*

As seen in the table below, this study employed employee engagement as a mediating variable to examine how safety climate and culture affected safety performance. The Sobel test is used to assess a mediating variable's influence and capacity to catalyze other well-established predictors and outcome variables. As a result, mediating effects happen when the Z-score is higher than 2.00 (absolute value) [77]. In this study, the Sobel test's Z-score value was greater than 2.00, as shown in Table 6. The other study variables were highly supported by the mediating impacts of employee engagement in their mediating role.

**Table 6.** Findings of the mediating effects of employee engagement.

| Mediation Effects | Coefficient | Standard Error | Sobel Test (Z-Score) | *p*-Value |
|---|---|---|---|---|
| SC→EEG→SP | 0.468 0.229 | 0.072 0.045 | 4.0069 | 0.000 |
| SCL→EEG→SP | 0.402 0.229 | 0.069 0.045 | 3.832 | 0.000 |

Note: SCL = safety climate; SC = safety culture; EEG = employee engagement; SP = safety performance.

*4.8. Analysis of Hypothesis Test Results and Decision*

This study's objective was to create and evaluate five hypotheses grounded in empirical and literature studies. The results also highlighted the importance of safety climate and employee engagement. The hypothesis test results indicated that safety culture had a positive and significant impact on safety performance, with an estimated value of 0. 0.511 *** and with $p = 0.000$, supporting Hypothesis H1. The hypothesis test results were used to examine these relationships. Safety culture, which encompasses shared values, beliefs, and attitudes toward safety within an organization, had a considerable and positive influence on safety performance. Likewise, the findings indicated that safety climate had a positive and significant effect on safety performance, with an estimated value of 0.236 *** and with $p = 0.000$), which supported Hypothesis H2. Employee engagement was found to have a positive and significant influence on safety culture, with an estimation value of $\beta = 468$ and with $p = 0.000$, as well as on safety climate, with typical estimation values showing a positive and significant influence ($\beta = 0.402$ **, $p = 0.000$), thereby supporting Hypothesis H3 and H4. Furthermore, the mediating variable, employee engagement, was found to have a direct and constructive impact on safety performance, with an estimated value of 0.229 *** and with $p = 0.001$, supporting Hypothesis H5.

This study highlighted the positive impact of safety culture and climate on safety performance in large-scale manufacturing enterprises in Ethiopia. It emphasized the importance of fostering a culture of safety where safety is valued and prioritized across an organization. This study also found that employee engagement positively impacted safety culture, climate, and performance, highlighting the importance of involving employees in safety-related activities. Overall, this study suggested that creating a safe workplace culture and engaging employees in safety-related activities can significantly improve organizational safety performance. The findings of all five hypothesis test results were confirmed, as demonstrated in Table 7.

**Table 7.** Summary of the anticipated results.

| Hypothesis | Paths | β | T-Value | *p*-Value | Decision |
|---|---|---|---|---|---|
| Hypothesis 1(+) | SC→SP | 0.51 | 10.63 | 0.000 | Supported |
| Hypothesis 2(+) | SCL→SP | 0.24 | 5.45 | 0.000 | Supported |
| Hypothesis 3(+) | SC→EEG | 0.47 | 6.51 | 0.000 | Supported |
| Hypothesis 4(+) | SCL→EEG | 0.40 | 5.83 | 0.000 | Supported |
| Hypothesis 5(+) | EEG→SP | 0.23 | 5.03 | 0.000 | Supported |

Note: SCL = safety climate; SC = safety culture; EEG = employee engagement; SP = safety performance.

## 5. Discussion

This study examined the effects of safety culture and climate on safety performance by mediating the impact of employee engagement in Ethiopian manufacturing enterprises. This study's results have significant global and regional implications, as they highlight the importance of safety culture, safety climate, and employee engagement in promoting workplace safety and improving safety performance in large-scale manufacturing enterprises. The findings are not only relevant to Ethiopia but are also applicable to other countries and regions facing similar challenges in ensuring workplace safety. This study emphasizes the crucial role of promoting a safety culture and providing a positive safety climate in improving safety performance. Additionally, this study highlights the importance of engaging employees in safety-related activities to enhance safety culture, safety climate, and safety performance. This discovery is consistent with prior research that underscores the significance of safety culture and climate in influencing deeds and enhancing safety performance [26,78,79].

Moreover, the results show that that employee engagement positively impacts safety culture, safety climate, and safety performance, and this highlights the critical role that employee engagement plays in shaping conduct and improving safety outcomes. Engaging employees in safety-related activities and fostering a sense of ownership and responsibility toward safety can motivate employees to adopt safe behaviors and prioritize safety. This study's findings suggest that improving employee engagement can positively impact safety performance. This finding aligns with previous research emphasizing the importance of employee engagement in enhancing safety performance [80]. The results provide valuable insights for organizations and policymakers seeking to improve workplace safety. They can inform the development of safety policies and strategies prioritizing safety culture, climate, and employee engagement.

This study also has economic and business sustainability impacts as it highlights the importance of workplace safety in promoting sustainable business practices. This study provides evidence of the causal relationships between safety culture, climate, safety performance, and the mediating role of employee engagement. These findings are crucial for organizations seeking to develop effective workplace safety programs that promote a culture of safety, which can ultimately lead to improved economic and business sustainability. By prioritizing workplace safety, organizations can reduce the costs associated with workplace accidents and injuries while also enhancing employee productivity and engagement. As such, this study provides valuable insights into the importance of workplace safety for economic and business sustainability [81,82].

In general, this study's results are helpful to policymakers and practitioners in developing strategies to create safer working environments in the manufacturing sector. It also supports the social cognitive theory.

Although this study offers a valuable understanding of the influences on safety culture, safety climate, employee engagement, and safety performance in Ethiopian manufacturing companies, there were some restrictions, such as its restricted scope and small sample size, that should be resolved in future research. By addressing these limitations, future research can expand on the results of this study and aid in developing a more comprehensive understanding of the intricate associations among safety culture, safety climate, employee engagement, and safety performance in manufacturing companies.

Drawing on this study's results, various recommendations for future research can enhance our comprehension of the effect of safety culture, safety climate, employee engagement, and safety performance on manufacturing companies. These include examining the moderating influence of organizational factors, examining the effects of safety interventions, and contrasting outcomes across industries and countries. These suggestions for further research can expand on the discoveries of this study and aid in developing a more comprehensive understanding of the associations among safety culture, safety climate, employee engagement, and safety performance in manufacturing companies [62,63,83,84].

**6. Conclusions**

This study investigated the impact of safety culture and climate on safety performance in Ethiopian large-scale manufacturing firms. Using employee engagement as a mediating factor, it employed a regression analysis, growth path modeling, and structural equation modeling with SPSS/AMOS. This study's findings suggested that safety culture, safety climate, and employee engagement are crucial in promoting a safe and healthy work environment. The results indicated that organizations should prioritize creating a safety culture and ensuring a safety climate to improve safety performance. Additionally, engaging employees in safety-related activities and fostering a sense of ownership and responsibility toward safety is essential.

Generally, this study's results have significant implications for understanding the factors that contribute to safety performance in the workplace. Employee engagement is critical, while safety culture and climate are essential factors in shaping safety performance. These insights can guide the development of effective workplace safety programs and promote a safety culture within manufacturing enterprises. This study's findings offer a valuable understanding of the elements that affect safety performance in the workplace and can assist in designing effective workplace safety programs with substantial and beneficial implications for theoretical research.

*6.1. Theoretical Implication*

The present study agreed with the social cognitive theory, which suggests that individuals acquire and shape their behavior through observation, modelling, and reinforcement. This study explored the link between safety culture, safety climate, employee engagement, and organizational safety performance. The findings supported the social cognitive theory that organizational factors like culture and climate shape individual behaviors and performances. Employee engagement was identified as a mediator that translates organizational influence into individual outcomes. This study also highlighted the direct impact of employee engagement on safety performance, which reinforces the role of intrinsic motivators in shaping desired behaviors. The integrated framework provided a systems perspective on occupational safety that aligned well with the social cognitive theory, favoring reciprocal interactions across levels. This study contributes theoretically and practically by developing and testing an integrated social cognitive framework and providing actionable recommendations for organizations.

*6.2. Managerial Implications*

This study's findings have significant managerial implications for large-scale manufacturing enterprises operating in Ethiopia. To improve organizational safety performance, managers should prioritize building and maintaining a strong safety culture that values safety and emphasizes employee responsibility. This can be achieved through clear communication, ongoing safety training, and regular discussions about safety-related issues. Additionally, managers should foster a positive safety climate by setting clear expectations, providing regular feedback, and implementing safety management systems. To enhance employee engagement, managers should focus on engaging employees in safety-related activities, providing opportunities for them to contribute, and involving them in decision-making processes related to safety. Managers should also look for opportunities to integrate

safety and employee engagement initiatives, monitor and evaluate their safety performance regularly, and continuously improve their organization's safety performance by adopting new strategies and learning from best practices. So, investing in safety culture, safety climate, and employee engagement can help organizations to achieve better safety outcomes and create a safer working environment. Generally, this study's findings suggest that creating a safety culture, ensuring a safety climate, and engaging employees in safety-related activities are critical components of improving safety performance in the workplace for large-scale manufacturing enterprises in Ethiopia.

### *6.3. Limitations and Future Directions*

This study was limited to only focusing on manufacturing enterprises. This study mainly tried to examine safety culture, safety climate, and safety performance; it did not use other constraints. This study was not deployed to other cross-sectional data considering other types of enterprises. Therefore, future research should emphasize other mediators, such as motivation, compliance, and safety participation.

**Author Contributions:** The authors of this manuscript are listed as F.L. and M.A. The roles and responsibilities of each author were as follows: conceptualization and methodology were contributed by both authors; software, validation, formal analysis, investigation, resources, and data creation of the initial draft preparation were primarily the responsibility of M.A.; F.L. contributed to writing the review, editing, and supervision. All authors have read and agreed to the published version of the manuscript.

**Funding:** This research received no external funding sources.

**Institutional Review Board Statement:** Not applicable.

**Informed Consent Statement:** Not applicable.

**Data Availability Statement:** The deployed data that support the findings of this research can be obtained from the corresponding author upon reasonable request.

**Acknowledgments:** First, we would like to thank the respondents who provided accurate data for our study. Furthermore, we thank the School of Management at the Wuhan University of Technology for their continuous support.

**Conflicts of Interest:** The authors declare no conflict of interest.

### Appendix A

**Table A1.** Measurement construct items.

| Variables | | Items | Source |
|---|---|---|---|
| Safety performance | (1) | Perceived importance of process safety information | [85,86] |
| | (2) | Perceived importance of operation procedure | |
| | (3) | Perceived importance of employee participation | |
| | (4) | Perceived importance of pre-start-up safety review | |
| | (5) | Perceived importance of overall incident investigation | |
| | (6) | Perceived importance of emergency planning | |

**Table A1.** *Cont.*

| Variables | | Items | Source |
|---|---|---|---|
| Safety culture | (1) | Perceived importance of safety rules | [2,8,87] |
| | (2) | Perceived importance of risk acceptance | |
| | (3) | Perceived importance of management commitment | |
| | (4) | Perceived effect of productivity pressure | |
| | (5) | Perceived importance of employee involvement | |
| | (6) | Perceived effect of individual responsibility | |
| Safety climate | (1) | Perceived management attitudes toward safety | [34,52] |
| | (2) | Perceived effects of safe conduct on promotion | |
| | (3) | Perceived status of safety officer | |
| | (4) | Perceived level of risk at the workplace | |
| | (5) | Perceived importance of safety training programs | |
| | (6) | Perceived importance of safety rules and procedures | |
| Employee engagement | (1) | Perceived importance of employee engagement in your work | [71,72,88] |
| | (2) | Perceived importance of commitment in your organization | |
| | (3) | Perceived importance of employee satisfaction in your job participation | |
| | (4) | Perceived recommendations for your organization to others as a good workplace | |
| | (5) | Perceived importance of employee engagement measurement and evaluation processes | |
| | (6) | Perceived importance of emergency planning for employee | |

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
