# Peer review of "The Influence of Safety Culture and Climate on Safety Performance: Mediating Role of Employee Engagement in Manufacturing Enterprises in Ethiopia"

_sustainability, doi:10.3390/su151411274_

Round 1
Reviewer 1 Report
Dear Authors,
Many thanks for the chance to review your manuscript. Please consider the following to help improve its quality and bring same to required standard.
General Comments:
1. The level of English is not good, and too many loose and out of context sentences exist.
2. It is very difficult to read and review a long manuscript poorly written. I would urge the authors to shorten the manuscript and make it cleaner and crisper.
3. More than 50% of references are numbered wrongly or out of place.
4. The discussion section requires more to the point, literature driven analysis.
5. Please attach the questionnaire used.
6. Refer the figures and table appropriately in the tex.t
7. Some statistical analyses like the one in Table 4 seems to have been done without proper explanation or reasoning, and seems to have been fitted to show more work.
Detailed Comments:
Introduction:
The author needs to be clearer and more succinct. 150 lines of the introduction section has so much of redundancy and hardly any context to Ethiopia and the topic relevancy in the African continent.
Line 36 – “how its personnel about them” refers to whom or what?
Line 45 – Needs more explanation that stating they are different
Line 47 – The sentence is discontinuous
Line 50 – Already mentioned in previous paragraph
Line 56 to 58 – Cite appropriate references
Lines 64 to 69 – Sentence is too vague. Reference 9 cited here does not talk about Ethiopian ministry report
Lines 70 – Have uniformity in referencing. Either use first author names or don’t. Accordingly, rewrite sentences throughout the manuscript.
Lines 73-76 – Ref12 does not talk about Norway study. Ref 13 does not talk about US study mentioned here
Lines 83-85 – The authors talk about their own result here. Please move it to discussion section.
Line 92 – Instead od “Some Researchers”, please cite appropriately
Lines 94-96 – Redundant lines. Same point discussed in lines 40’s
Lines 100-104 – Again referring to own study inference. Seems like authors have not proof read before submitting
Lines 107-112 - Please cite the source
Line 109 – When abbreviations are used for first time, please expand.
Line 118 – Ref 18 is talks about environmental toxicology. This shows the level of carelessness in proofreading
Lines 130-134 – Again referring to own study inference. See previous comments
Lines 133-137 – Please move it to discussion
Literature Review and Hypothesis Development:
330 lines is too much and the authors could easily explain and arrive at the hypotheses in less than 100 lines of text. Just citing papers is not literature review. The authors need to link it efficiently with the context and draw the reader to the point.
Line 153 – Sentence not clear, please rephrase
Lines 165-167 – You cannot rewrite sentences from blog sites and that too as direct quotes without giving proper sources
Lines 169 – Who are the ‘Others”?
Lines 174 to 176, 178 to 182, 183 to 184, 184 to 186 – Why are these quotes here? Please rewrite and cite appropriately with the connect of significance
Line 196 to 198 – Please cite more references as the author indicates “A growing body of research”
Line 209, Lines 231 to 233, Line 244, Line 324 – Maintain uniformity in citations
Line 215 – Both of the cited papers are not Christian et al.,
Lines 259 to 279 – Reference numbering 43 to 46 is missing.
Line 312 to 313 – All the cited literature still does not make it clear why the authors arrive at this hypothesis. Instead, they can just say these are our hypotheses and we want to test them.
Line 320 – Please cite the “One Study”
Lines 355 to 356 – Sentence incorrect. Rewrite
Line 383 – Discuss the debates. That’s the whole idea of lit review. This will also give clarity on why you arrived at the hypotheses
Line 383-388, Line 428 – I don’t see the relation to Ethiopia
Line 413 – Cite the many findings. These kinds of phrases make it so vague to read out
Line 437 – Its is reference number 67. Follow uniformity in format of citation
Lines 478 to 483 – Please clearly distinguish business performance, safety performance and firm performance in here. Also please bring in the context why Ethiopia?
Line 485 – Figure 1 not referred in text.
Research Methodology:
Line 495 – Ref 72 and 73 does not discuss about Ethiopian industries
Line 504 – Ref 74 does not talk about sampling techniques
Line 507 – The authors mention “primary and secondary data” apart from the questionnaire response. Please elaborate on what are these?
Line 502-510 – Please give the questionnaire as supplement and refer it here. Also give the ethical clearance information for the questionnaire used
Results and Extrapolations:
Line 523 to 529 – Please give these demographics in the form of a table
Lines 536-537 – How does the author arrive at this conclusion?
Line 542 – Expand and Define AVE
Line 552 to 553 – What are the significance of these calculated entities. Please discuss it in a few lines or give references if applicable.
Lines 556-557 – Same as previous sentences. Try avoiding redundancy.
Line 560 – How does the authors say factor loading of 0.70 is significant. Give some basis for it.
Line 567 – Ref 79 talks about Corporate Social Responsibility. How is it related to statistics and data analyses?
Line 573 – In Table 1, what is 1 to 5 in item column refer. Are they iterations or replicate analyses?
Lines 583 to 585 – Isn’t the outcome variables same as mediation variables?
Line 612 – Table 3. Please give “n” as the degrees of freedom. Otherwise, the obtained correlation values are meaningless.
Line 616 – The authors say “researchers likely used”. Why are the authors not sure of what they did? If not sure, please remove it.
Line 631 – Table 4 – Why do the authors say 0.90 is the recommended value for GFI, and other indices. Isn’t 95% is what is used to test significance. Going by that, all indices show the tested model is insignificant. Better remove the table.
Also, the degrees of freedom are 110 in the table, while the respondents for the questionnaire is 368. Why this discrepancy?
Lines 649-651 – What is the basis for this conclusion?
Lines 664-667 – Not sure what is conveyed. Please rephrase.
Line 683 – It’s not clear how these values were arrived here. Explain the methodology in a better way.
Line 713-728 – A lot of redundant sentences. Can be made into a short paragraph of 5-6 lines
Line 732 – Please give info on the number of samples. Significances are meaningless without it.
Discussion:
This section needs to be much more elaborated. It completely lacks global and regional significance.
Line 739, 744 – The authors use words like “particularly attractive” and so on, referring to their results. This shows a lot of bias. Please remove such phrases.
Ref 81, 82, 83 are all references from studies in China. This study is done in Ethiopia, so give more global and regional references.
Line 753-758 – Highlighting the methodology of your study without any referencing to relevant studies cannot be considered as discussion. Please remove these lines
Author Response
Dear. Reviewer,
We would like to express our sincere gratitude for taking the time to read and review our article. your insightful comments and questions have been invaluable in helping us improve the quality of our work. our response for these questions and comments has attached as follows. Thank u

Reviewer 2 Report
The paper investigated the influence of safety culture and climate on safety performance. The research content has certain theoretical and practical significance.There are several suggestions for further improvement:
1. The introduction section should reflect the relationship between the research and sustainability.
2. Why are the third and forth hypotheses using "significant impact", and the fifth hypothesis using "substantial impact", however the first two hypotheses not using" significant nor substantial influence?".Please give more explanations.
3. It is recommended to upload the survey questionnaire as an attachment for reference.
4. The analysis of the paper uses structural equations, and it is recommended to provide a brief introduction to the theoretical basis of structural equations.
5.Suggest explaining the differences in characteristics among survey participants。
It is recommentd to resubmit the paper after major revisions.
Author Response
Dear, reviewers
Thank you for your comments and questions and we try to answer all the questions as follows in the attached document

Reviewer 3 Report
Please see atached comments

Author Response
Dear, reviewer
Thank you for your valuable questions and suggestions hence wet tried to answer all questions and suggestions as follows

Reviewer 4 Report
The paper - The Influence of Safety Culture and Climate on Safety Performance: Mediating Role of Employee Engagement in Manufacturing Enterprises, Ethiopia's, is interesting, however, some revisions are needed.
The paper needs to be language check – for instance within the abstract: The study uses a quantitative research methodology, and purposive random sampling is used to select 368 respondents from five large-scale industrial manufacturing enterprises, including employees surveyed.
Perhaps - Using a quantitative research methodology, 368 respondents from five large-scale industrial manufacturing enterprises were selected through purposive random sampling.
Line 69 -70, Moreover, nowadays, safety in manufacturing enterprises is critical there is a different
interpretation of safety, according to Zhanming Li et. Al.
should be “et al.” – this statement needs to be rephrase.
Perhaps - Additionally, according to Li et al. (citation also needed), safety in manufacturing enterprises has become increasingly critical in recent years.
Typing errors – line 524 – “workers.182 (49.6)” use similar convention as in 50.4% (186)
Many sections need to be recheck for proper citations – lines 34 to 35 - "Safety climate" and "safety culture" is frequently used in the literature to describe how a business handles safety-related issues and how its personnel about them.
Would suggests to provide more detailed figure 1, including the hypotheses (which can be signify by H1 … etc.
Did you test for skewness and kurtosis? How about normality of the data? How about bootstrapping? heterotrait-monotrait ratio of correlations of the variables/factors?
Figure 2 – should be redraw – should use similar names for the factors – see
Schreiber, J. B., Stage, F. K., King, J., Nora, A., & Barlow, E. A. (2006). Reporting structural equation modeling and confirmatory factor analysis results: A review. Journal of Educational Research, 99(6), 323-337. https://doi.org/10.3200/JOER.99.6.323-338
Research ethics? Consent?
Control variables? Were they considered if not why? if yes, should also be noted.
Lines 524, According to statistics, women participate in society in a largely positive manner; this might result from LSME's capacity to consider the participation of female employees. – should be cited. Not clear on the purpose of this statements.
In sum, the paper is somewhat interesting, however, the presentation of the paper makes it hard to read, many sections need citation. Presentation of statistical data should be revisit. Paper needs to be language check.
The paper needs to be language check – for instance within the abstract: The study uses a quantitative research methodology, and purposive random sampling is used to select 368 respondents from five large-scale industrial manufacturing enterprises, including employees surveyed.
Perhaps - Using a quantitative research methodology, 368 respondents from five large-scale industrial manufacturing enterprises were selected through purposive random sampling.
Line 69 -70, Moreover, nowadays, safety in manufacturing enterprises is critical there is a different
interpretation of safety, according to Zhanming Li et. Al.
should be “et al.” – this statement needs to be rephrase.
Perhaps - Additionally, according to Li et al. (citation also needed), safety in manufacturing enterprises has become increasingly critical in recent years.
Typing errors – line 524 – “workers.182 (49.6)” use similar convention as in 50.4% (186)
Author Response
Dear, Reviewer,
Thank u for all your valuable questions and suggestions and we tried to responses as follows.

Round 2
Reviewer 1 Report
Dear Authors
Many thanks for taking time to respond to my earlier comments. I still find the manuscript needing further update. Refer to these comments below;
i. The introduction chapter is unnecessary long and this will tend to deviate/dissuade yoir target audience from engaging with the main aspect of the work. In addition, the you have also considered a lengthy literature review together running to around 11pages. This need looking at to help keep scene setting simple and concise
ii 3.2 lacked clarity and will advise this also need revising
iii. 3.3 line 535-536 did not come out clear. Please revise. You also stated questionnaire was altered. Not sure what you meant here?
iv. it will be great to report the breakdown of the repsondent based on the number of firms they were drawn as there was vague statement regard the location house major/large manufacturing firms
v. line 518, it was not clear the phase "they". You need to be clear with the use of such words which are dotted round the paper
vi. Conclusion is unnecessary long this need be kept simple and clear. Possibly move certain points to the discussion/result where possible.
vii. Overall, the manuscript require further proofread
Hope this comment will help improve the work further
Regards
Author Response
Dear, reviewer my second round of responses has attached below.

Reviewer 2 Report
All the comments have been addressed. I am satisfied with the revised manuscript. The authors should do a thorough proofreading of the entire manuscript before publication to avoid unnecessary wording and grammatical issues.
Author Response
Thank you so much for your feedback.

Reviewer 4 Report
Just a minor concern about informed consent, this should be clarify within the research methodology.
just some minor typing error and grammar check needed
Author Response
Thank so much.

Round 3
Reviewer 1 Report
Dear Authors,
Many thanks for taking time to respond to my observation.
I recommend the manuscripts be accepted in its present format.